Microbiology **Spectrum**

ⓐ | **Open Peer Review** | Host-Microbial Interactions | Research Article

# Study on geographic differentiation and environment-host synergistic assembly mechanism of root-associated fungal communities in *Paphiopedilum purpuratum*

Yong Tan,[1,2] Junxi Liang,[1,2,3] Qifei Yi[1,2]

**ABSTRACT** The orchid–fungus symbiosis is fundamental to orchid survival and reproduction; however, the diversity patterns and assembly mechanisms of the root-associated mycobiota in *Paphiopedilum purpuratum* remain inadequately characterized. We utilized high-throughput sequencing of the internal transcribed spacer 2 region to investigate the composition, diversity, sources, and assembly processes of the endophytic fungal communities across eight geographically distinct populations, with complementary profiling of rhizosphere soil fungi. Our results indicated that Ascomycota constituted the dominant phylum within the root mycobiota, while core taxonomic groups exhibited pronounced geographic differentiation at both family and genus levels. Significant inter-population disparities in α-diversity metrics reflected underlying community compositional divergence. Environmental variables, particularly longitude, exerted a stronger influence on community structure than biotic factors. Approximately 44.05% of root fungal operational taxonomic units were soil-derived, and the host plant selectively enriched fungal taxa, most of which possessed unknown trophic modes. Community assembly processes were compartment-specific: the root endophytic mycobiota was primarily governed by stochastic ecological drift, whereas the rhizosphere communities were predominantly shaped by deterministic dispersal limitation. This compartment-specific assembly was evidenced by the prevalence of stochastic processes ($|\beta NTI| < 2$) in the root endosphere, contrasting with the dominance of deterministic processes ($|\beta NTI| > 2$) in the rhizosphere. Co-occurrence network analysis revealed higher connectivity and robustness in the endophytic mycobiota. The interaction network between orchid mycorrhizal fungi and other root-associated soil fungi formed an efficient and stable functional system whose complexity showed population-specific differentiation. Collectively, our findings demonstrate clear geographic divergence in the root fungal communities of *P. purpuratum* and underscore a synergistic environment–host assembly mechanism, thereby providing critical ecological insights for informing conservation strategies for this endangered orchid.

**IMPORTANCE** This study investigates the root-associated fungal communities of the endangered orchid *Paphiopedilum purpuratum* across its geographical distribution. We identified clear geographical differentiation in community composition and diversity, predominantly driven by abiotic factors—particularly longitude—rather than biotic factors. A key finding reveals that 44% of root fungal taxa originate from the soil, indicating active host-mediated selection. A fundamental dichotomy in assembly mechanisms was observed: stochastic ecological drift dominated within roots, whereas deterministic dispersal limitation prevailed in the rhizosphere. Co-occurrence networks demonstrated that the root fungal community is highly connected and robust, suggesting a stable functional system. Our findings elucidate the synergistic roles of environment and host in shaping fungal assembly, providing novel insights into orchid–

**Peer Reviewer** Youwei Zuo, Southwest University, Chongqing, China

Address correspondence to Qifei Yi, yiqifei@scbg.ac.cn.

The authors declare no conflict of interest.

fungus symbiosis with theoretical implications for mycorrhizal ecology and practical relevance for conservation strategies.

**KEYWORDS** *Paphiopedilum purpuratum*, root-associated fungi, fungal diversity, community assembly, co-occurrence network, ecological drift, orchid mycorrhiza, microbial ecology

The escalating loss of biodiversity poses a significant threat to global ecological security, driven primarily by habitat fragmentation and resource overexploitation (1). While sustainable development goals seek to reconcile conservation with economic growth, inherent conflicts often persist between wildlife protection and human activities (2). International frameworks like Convention on International Trade in Endangered Species of Wild Fauna and Flora (CITES) and the Convention on Biological Diversity (CBD) have demonstrated that effective conservation can sustain ecosystem services while balancing ecological and human needs, thereby providing a critical foundation for the *in situ* conservation of endangered species. *Paphiopedilum purpuratum* (Lindl.) Stein, a flagship orchid species, is now endangered due to habitat fragmentation, over-collection, and climate change. Its wild populations persist in small, isolated patches and are classified as Endangered (EN) on the International Union for Conservation of Nature Red List. *In situ* conservation is essential to reverse this trend, as it maintains ecosystem integrity, supports natural regeneration, and facilitates the sustainable use of genetic resources (3, 4). However, targeted *in situ* strategies for *P. purpuratum* are currently lacking, and their development urgently requires multidisciplinary evidence, particularly concerning the role of mycorrhizal fungi. Orchids depend entirely on mycorrhizal symbiosis for germination, growth, and reproduction. They recruit specific orchid mycorrhizal fungi (OMF) to obtain essential nutrients and enhance stress resilience (5, 6). These fungi colonize roots or reside in the soil, forming complex networks that directly influence host population dynamics (7, 8). Through root exudates, plants selectively enrich beneficial OMF from the soil microbial pool (9, 10), making an understanding of these associations indispensable for orchid conservation. Although mycorrhizal research has advanced for orchids like *Dendrobium* and *Cymbidium* (11, 12), including insights into how environmental factors shape fungal diversity, specificity, and adaptability (5, 13), the mycorrhizal ecology of *P. purpuratum* remains largely unexplored. Critical knowledge gaps include (i) the mechanisms underlying the geographic differentiation of its root-associated fungal communities (14), (ii) the quantitative impact of environmental factors on these communities (15), and (iii) the sources of root-associated fungi and the structure of their co-occurrence networks (16). Existing studies on *P. purpuratum* have focused on tissue culture (17), community structure (18), and floral morphology (19), leaving a substantial void in understanding mycorrhizal community assembly, environmental responses, and host–fungal interactions. This gap critically impedes the formulation of effective protection strategies. In natural environments, a thorough understanding of how the mycorrhizal fungi of *P. purpuratum* respond to environmental gradients and interact with the broader soil fungal community is fundamental for habitat restoration, population expansion, and successful reintroduction. This study investigates eight wild populations across Guangdong and Fujian provinces, China. By integrating high-throughput sequencing (internal transcribed spacer 2 [ITS2] region), multivariate statistics, and ecological modeling, we aim to (i) characterize the root-associated fungal communities, (ii) quantify the influence of key environmental drivers, and (iii) elucidate the underlying interaction network mechanisms. Our findings will establish a theoretical basis for the conservation of *P. purpuratum* and provide essential mycorrhizal ecological evidence to support its *in situ* conservation and habitat restoration, thereby enhancing *ex situ* preservation efforts for this endangered orchid.

## MATERIALS AND METHODS

### Sampling of plants and soil

Sampling was conducted across eight natural populations of *P. purpuratum* located in Guangdong and Fujian provinces, China (Fig. 1). Within each population, sampling was performed in typical areas with relatively high plant density. Healthy plants were randomly selected using sterile gloves. Using aseptic techniques, one to two segments of root tissue (approximately 3–5 cm in length) were collected from each plant, ensuring post-collection survival, and the corresponding rhizosphere soil was simultaneously gathered. For low-density populations, sampling intervals were maintained at >50 cm, whereas for high-density populations, the interval was increased to >1 m. Root samples from two to three individual plants were pooled to form one biological replicate. Each sampling site included three biological replicates for both root and rhizosphere soil samples. All samples were stored in 50 mL sterile centrifuge tubes, transported to the laboratory on ice, and subsequently stored at –80°C until DNA extraction. Additionally, approximately 500 g of bulk soil was collected from each site for physicochemical analysis. Root segments were initially rinsed with sterile distilled water to remove adhering soil and then subjected to surface sterilization in a laminar flow hood. The sterilization procedure was as follows: immersion in 75% ethanol for 30 s, followed by three rinses with sterile distilled water; treatment with 20% sodium hypochlorite for 5 min; blotting on sterile filter paper; and multiple final rinses with sterile distilled water. To verify the effectiveness of the surface sterilization, 500 µL of the final rinse water was plated onto Potato Dextrose Agar medium and monitored for microbial contamination.

All root samples were subjected to amplicon sequencing targeting the ITS2 region. Soil samples were sequenced for both the ITS2 region and the 16S rRNA gene. A total of 72 samples (including roots and rhizosphere soil) from the eight sites were included in the subsequent analyses. It should be noted that while three biological replicates per population represent a standard preliminary approach in ecological studies for capturing major community trends, we fully recognize that a larger sample size would enhance statistical power, particularly for resolving fine-scale geographic differentiation patterns. The sample size in this study was determined based on the availability of representative individuals across different geographic regions and was designed to provide initial

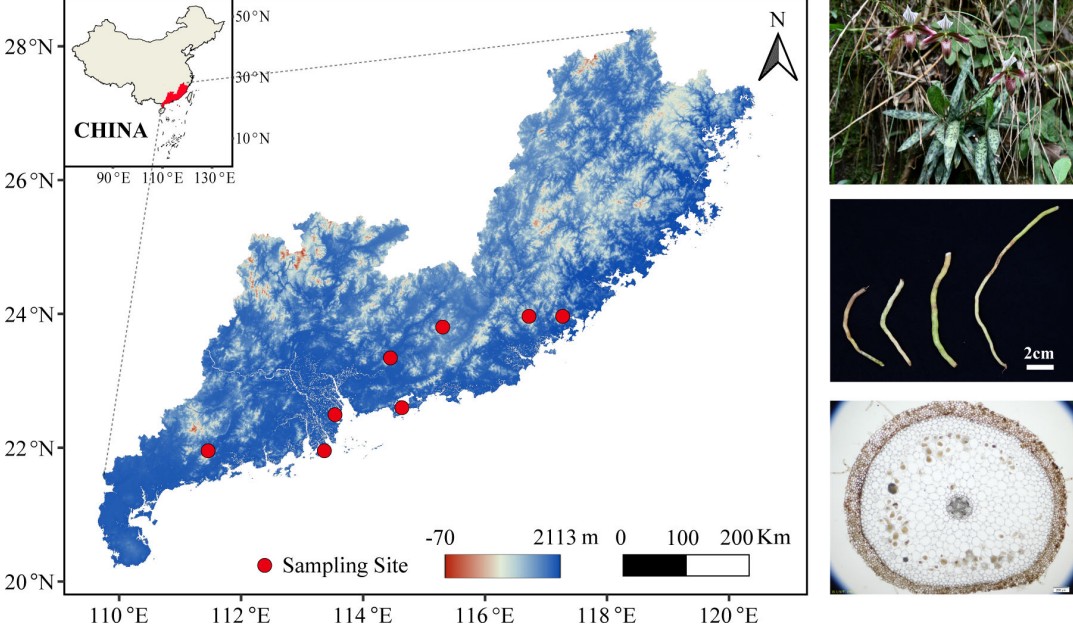

**FIG 1** Sampling site information of *P. purpuratum* populations. A total of eight populations were distributed mainly in Guangdong province and Fujian province of China. The three figures on the right, from top to bottom, are wild habitat, root, and root cross-section.

insights into the assembly mechanisms of root-associated fungal communities along a geographical gradient. Future studies with expanded sample sizes are warranted to validate and refine the preliminary findings presented here.

## Acquisition of environmental factors

Soil physicochemical properties, including total nitrogen (g/Kg), total phosphorus (g/Kg), total potassium (g/Kg), total organic carbon (g/Kg), organic matter (g/Kg), pH, available phosphorus (mg/Kg), available potassium (mg/Kg), ammonium nitrogen ($NH_4^+$ mg/Kg), and nitrate nitrogen ($NO_3^-$ mg/Kg), were analyzed using standard laboratory procedures. Climate variables—namely, elevation (Elev m), annual precipitation (Prec mm), wind speed (Wind m s$^{-1}$), vapor pressure (Vapr Kpa), solar radiation (Srad kJ m$^{-2}$ day$^{-1}$), and annual mean temperature (°C)—were obtained from the WorldClim database (https://www.worldclim.org/) at a 30-arcsecond resolution. Additionally, the aridity index and evapotranspiration data were sourced from a published figshare repository [20]. All climatic variables were extracted for each sampling location using the raster package (v3.6-32) in R. Biotic factors were derived from cross-domain (inter-kingdom) network analysis, which integrated rhizosphere bacterial, root endophytic fungal, and rhizosphere fungal communities. Key topological metrics—including the average degree (Degree), the ratio of negative associations (c_neg), and the ratio of inter-domain associations—were calculated from these networks and included as biotic variables. We incorporated these metrics for two primary reasons: first, rhizosphere microbial communities respond to environmental fluctuations and collectively influence plant physiology; second, complex interactions among rhizosphere microbes can directly or indirectly (via plant-mediated mechanisms) shape the assembly of root-associated fungal communities. Thus, this cross-domain approach allows for the elucidation of the complex inter-kingdom interactions that govern root mycobiome assembly.

## DNA extraction and high-throughput sequencing

Total genomic DNA was extracted from all samples using the cetyltrimethylammonium bromide/sodium dodecyl sulfate method. The concentration and integrity of the extracted DNA were assessed by 1% agarose gel electrophoresis, and samples were diluted to 1 ng/µL with sterile water. Amplification of the bacterial 16S rRNA gene V4–V5 hypervariable regions was performed using the barcoded primers 515F and 907R, while the fungal ITS1 region was amplified with the barcoded primers ITS1F and ITS2R. Polymerase chain reaction (PCR) was carried out in a 30 µL reaction mixture containing 15 µL of Phusion High-Fidelity PCR Master Mix (New England Biolabs), 0.2 µM of each forward and reverse primer, and approximately 10 ng of template DNA. The thermal cycling conditions consisted of initial denaturation at 98°C for 1 min; followed by 30 cycles of denaturation at 98°C for 10 s, annealing at 50°C for 30 s, and extension at 72°C for 60 s, with a final extension at 72°C for 5 min. The PCR products were verified by 2% agarose gel electrophoresis. Amplicons of the expected size (approximately 400–450 bp) were pooled in equimolar ratios and purified using the GeneJET Gel Extraction Kit (Thermo Scientific). Sequencing libraries were constructed with the NEBNext Ultra DNA Library Prep Kit (New England Biolabs, USA), and library quality was assessed using the Qubit 2.0 Fluorometer and Agilent Bioanalyzer 2100 system. The pooled libraries were sequenced on an Illumina MiSeq platform to generate 250/300 bp paired-end reads. Raw sequencing reads were processed using FLASH for read merging and demultiplexing based on sample-specific barcodes. UPARSE was used to cluster the quality-filtered sequences into operational taxonomic units (OTUs) at a 97% similarity threshold. Representative sequences from each OTU were taxonomically classified using the RDP classifier.

## Statistical analysis

All statistical analyses and visualizations were performed in R (v4.5.0). Microbial feature tables, taxonomic assignments, and sample metadata were integrated and managed

using the phyloseq package (v1.52.0) (21). Multi-omics data sets, including bacterial 16S rRNA gene and fungal ITS amplicon data, were merged into a unified object using the merge_phyloseq function. Taxonomic aggregation at the phylum, family, and genus levels was conducted via the tax_glom function. Alpha diversity was assessed by calculating the Shannon, Simpson, and Chao1 indices using the diversity function in the vegan package (v2.6-10) (22). Significant differences between groups were assessed by one-way analysis of variance (ANOVA) followed by Tukey's honestly significant difference *post hoc* test ($P < 0.05$), with results denoted by lowercase letters (e.g., a, b, and ab) in the figures. Beta diversity was evaluated using Bray–Curtis dissimilarity. Principal coordinate analysis was performed using the cmdscale function, and permutational multivariate analysis of variance with 999 permutations was conducted using the adonis2 function in vegan to test for group differences. Distance-based redundancy analysis (dbRDA) was carried out using the rda function, and significant environmental variables ($P < 0.05$) were selected via ANOVA with stepwise model selection. Variation partitioning analysis (VPA) was implemented using the varpart function to quantify the independent and joint contributions of abiotic and biotic factors to the community variation. Phylogenetic analysis of dominant fungal taxa was conducted by constructing neighbor-joining trees with the nj function in the ape package (v5.8-1) (23), supported by 1,000 bootstrap replicates. The ggtree package (v3.7.23) (24) was used for tree visualization with overlaid heatmaps. Random forest models were trained using the randomForest package (v4.7-1.2) (25), and the top 1% of OTUs ranked by MeanDecreaseGini were retained as dominant predictors. Functional annotation of fungi was performed using the FUNGuild database via the funguild_assign function (v0.3.0) (26). Microbial source tracking was conducted using the SourceTracker package (27), and differences among source groups were tested using analysis of similarities in vegan. Differential abundance analysis was performed using the EnhancedVolcano package (v1.26.0) (24) with thresholds set at $|\log_2 (FC)| \geq 1$ and $P < 0.05$. Spearman correlations between taxa and environmental variables were computed and visualized using the corrplot package, with significance levels denoted as * $P < 0.05$, ** $P < 0.01$, and *** $P < 0.001$. Community assembly processes were assessed using the neutral community model via the neutral community model package (28), with model fit evaluated by $R^2$ and dispersal parameter $N_m$. Stochasticity and determinism were further quantified using a null model approach with 999 permutations via the nullmodel function in vegan, and the Raup–Crick metric was applied to distinguish ecological processes. Co-occurrence networks were inferred using the SparCC algorithm in the SpiecEasi package (v1.1.3) (28). Associations with correlation $\geq 0.6$ and $P < 0.05$ were retained. Topological properties including node degree and betweenness centrality were computed using the igraph package (v2.1.4) (29). Cross-domain network visualization was performed using the ggClusterNet package (v2.0.0) (30), and final layouts were refined in Gephi. Network robustness and module cohesion were evaluated using the meconetcomp package by simulating node removal and quantifying changes in connectivity.

## RESULTS

### Characteristics of community composition and α-diversity differentiation

Although Ascomycota dominated the root mycobiota of *P. purpuratum* at the phylum level (Fig. 2A), family and genus-level compositions exhibited clear geographic differentiation (Fig. 2B and C), indicating that community assembly is shaped by local availability and environmental conditions rather than host-specific selection. This geographic structuring was paralleled by significant α-diversity disparities among populations (Fig. 2D through F): GDCZ showed higher Shannon and Simpson indices than GDSZ and a higher Simpson index than GDHY, while GDMM had greater species richness (Chao1) than GDHZ. The concomitant regional divergence in taxonomy and α-diversity implies that community structure is molded by the combined action of stochastic and deterministic assembly processes.

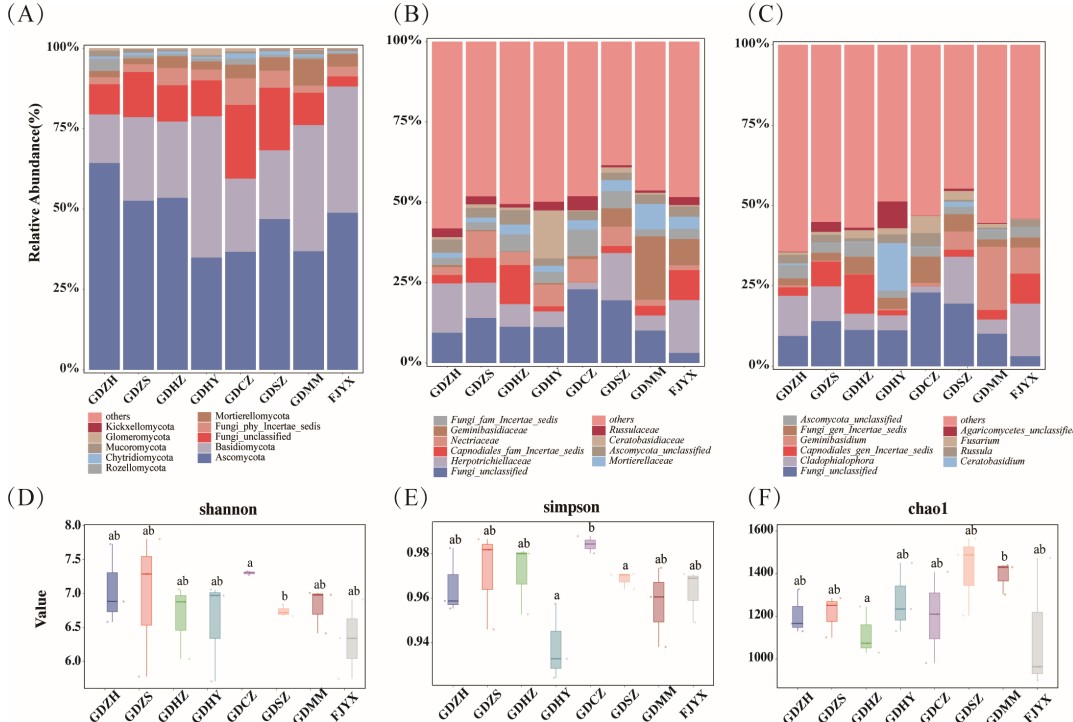

**FIG 2** Taxonomic composition and α-diversity of root-associated fungal communities across populations. (A–C) Relative abundance of fungal taxa at the phylum (A), family (B), and genus (C) levels, illustrating the dominance of Ascomycota and geographic variation in core taxa among populations. (D–F) Boxplots of α-diversity indices: Shannon (D), Simpson (E), and Chao1 (F). Significant differences among populations were assessed by one-way ANOVA with Tukey's *post hoc* test ($P < 0.05$). Groups not sharing the same lowercase letter (e.g., a, b) differ significantly, reflecting population-level divergence in species richness and evenness.

## Community structure differentiation and environmental driving factors

Fungal communities associated with *P. purpuratum* roots exhibited significant population-level structure, as shown by principal component analysis (PCA) ($P < 0.001$; Fig. 3A). The clear separation of populations, particularly FJYX, points to the strong imprint of spatial and environmental gradients. This was quantitatively confirmed by dbRDA, in which environmental variables explained 46.9% of community variance, with longitude being the most significant factor (Fig. 3B). The overwhelming influence of the abiotic environment was cemented by VPA, which assigned 30.52% of explained variance to abiotic factors alone, dwarfing the non-significant role of biotic factors (Fig. 3C). Finally, random forest analysis identified solar radiation and precipitation as the key abiotic filters determining the composition of the core fungal taxa (Ascomycota and Basidiomycota; Fig. 3D). This multi-method convergence firmly establishes environmental filtering as the dominant assembly mechanism.

## Functional group differentiation characteristics of core OTUs

Core mycobiota analysis identified 355 persistent OTUs shared across all populations (Fig. 4A), dominated by unannotated families, *Herpotrichiellaceae* and *Nectriaceae* (Fig. 4B). Functional profiling revealed (i) endophytic fungi were significantly enriched in GDCZ, GDSZ, and GDMM populations ($P < 0.05$); (ii) OMF remained scarce except in FJYX where they were significantly more abundant ($P < 0.05$) (Fig. 4C). Network analysis identified 373 keystone OTUs with distinct functional specialization: (Fig. 4D) while persistent OTUs were enriched in pathogens and endophytes, keystone OTUs showed significant OMF enrichment ($P < 0.01$) (Fig. 4E and F). This demonstrates that fungi with high network connectivity are preferentially symbiotic, whereas the broader persistent community represents generalist colonizers.

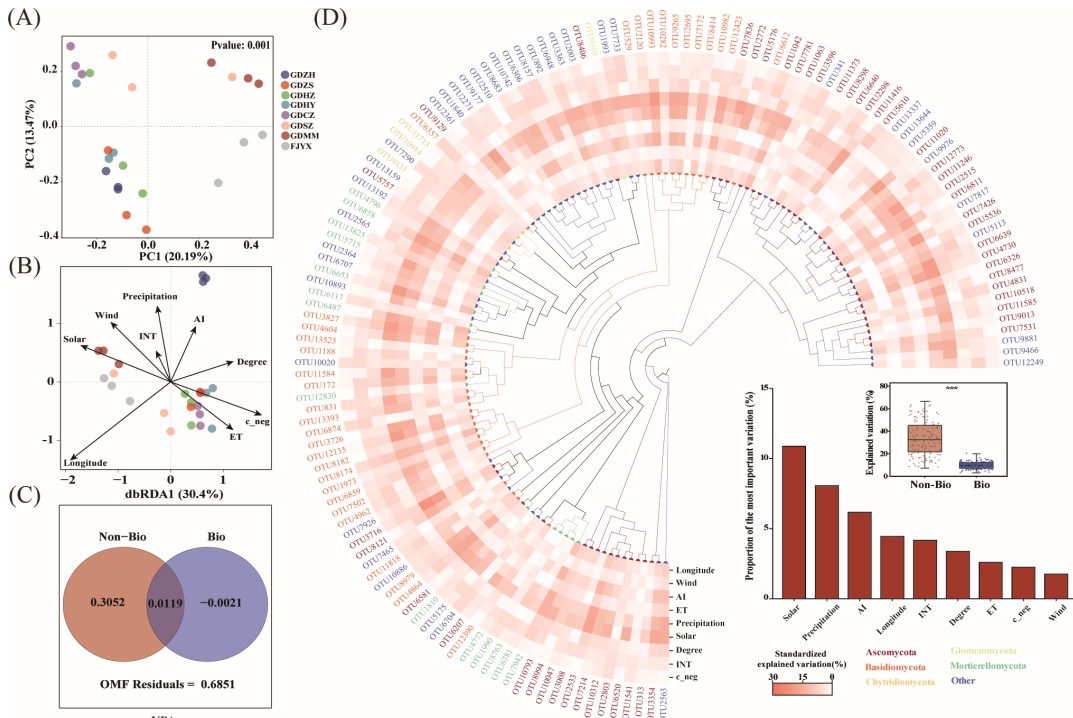

**FIG 3** β-diversity and assembly mechanisms of root-associated fungal communities across eight geographical populations. (A) PCA reveals significant population-level segregation along the first two axes, which collectively explain 33.63% of the total community variation (*P* < 0.001). (B) dbRDA shows that environmental variables explain 46.9% of the community variation, with longitude exhibiting the strongest influence. (C) VPA indicates that abiotic factors account for 30.52% of the explained variation, substantially exceeding the contributions of biotic and spatial factors. (D) Phylogenetic and random forest identify solar radiation and precipitation as the key environmental predictors shaping the composition of the top 1% dominant fungal taxa.

## Source characteristics of root-associated fungi and differences in community assembly mechanisms

Comparative analysis with rhizosphere soil fungi revealed that 44.05% of root-associated fungi in *P. purpuratum* were soil-derived, while 55.95% originated from unknown sources (Fig. 5A). The proportion of soil-derived fungi showed significant inter-population variation (Fig. 5B), indicating geographical heterogeneity in fungal recruitment. Through integrated Venn diagram and differential abundance analysis, we identified 235 OTUs that were significantly enriched in roots compared to soil (Fig. 5C and D), with their abundance exhibiting clear population-level differentiation (Fig. 5E).

Functional annotation revealed that root-enriched fungi were dominated by taxa of unknown ecological function, followed by multi-trophic fungi, saprotrophs, and symbiotrophs (Fig. 5F). The relative abundance of symbiotic fungi was highest in GDCZ and lowest in GDHZ, suggesting potential host-environment interactions in recruitment.

A significant negative correlation between the β-nearest taxon index (βNTI) of root-associated and rhizosphere soil fungi indicated contrasting community assembly processes (Fig. 5G). Among root-associated fungi, only GDSZ was governed by dispersal limitation, while the other seven populations were dominated by ecological drift (Fig. 5H). In contrast, rhizosphere communities in six populations were structured by dispersal limitation, with only GDMM and FJYX driven by ecological drift (Fig. 5I). Overall, root-associated fungal assembly was predominantly influenced by ecological drift with secondary dispersal limitation, while the opposite pattern prevailed in the rhizosphere (Fig. 5J).

Neutral community model analysis confirmed stronger stochastic influences in root-associated communities (55.9% variance explained) compared to the rhizosphere (47.2%, Fig. 5K and L), with rhizosphere fungi exhibiting significantly higher dispersal capacity.

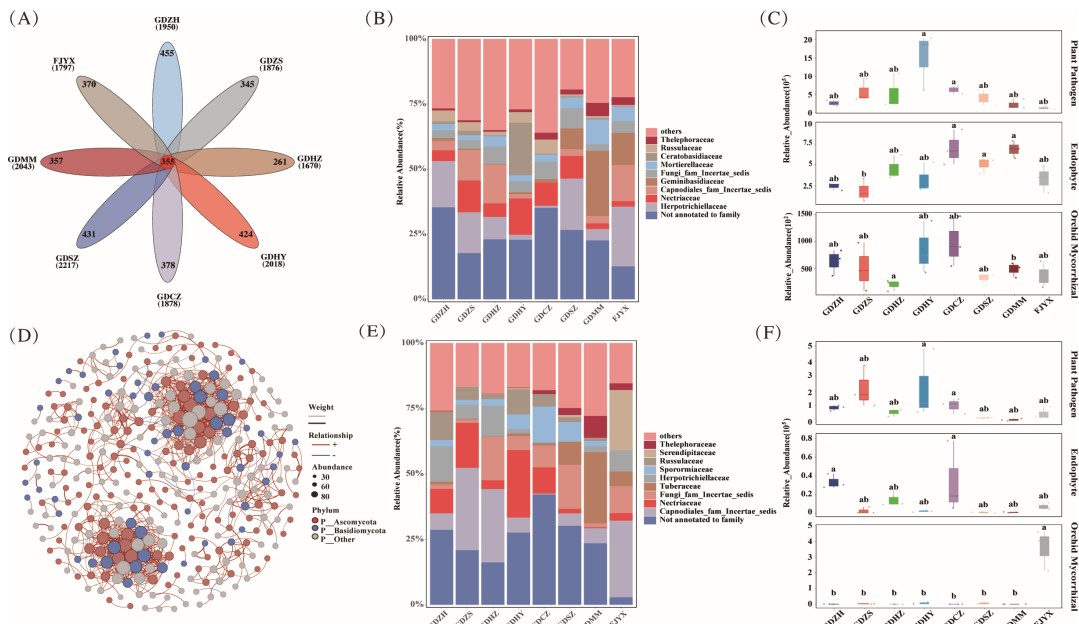

**FIG 4** Identification and functional characterization of core root-associated fungi through persistent and keystone taxa analysis. (A) Venn diagram identifies 355 persistent OTUs shared across all 8 populations. (B) Taxonomic composition of persistent OTUs at the family level, dominated by unannotated families, *Herpotrichiellaceae* and *Nectriaceae*. (C) Functional guild abundances of persistent OTUs across populations. Letters indicate significant differences (one‑way ANOVA, Tukey's test, $P < 0.05$); endophytes are more abundant in GDCZ, GDSZ, and GDMM than in GDZS, whereas OMF remain low overall. (D) Co‑occurrence network reveals 373 keystone OTUs (degree >1). (E) Taxonomic profile of keystone OTUs at the family level, showing dominance by unannotated families, *Capnodiales fam Incertae sedis* and *Nectriaceae*. (F) Functional guild distribution of keystone OTUs. OMF are significantly more abundant in FJYX than in other populations. Comparative analysis indicates that OMF are markedly enriched in keystone OTUs, whereas plant pathogens and endophytes are more abundant in persistent OTUs.

## Network differentiation between root-associated and rhizosphere soil fungi

Network robustness analysis revealed that the root-associated fungal (OMF) network exhibited significantly higher stability than the rhizosphere soil fungal (other root-associated soil fungi [ORSF]) network ($P < 0.05$). Under progressive node removal simulations, the OMF network maintained superior performance in both global efficiency and eigenvector centrality metrics (Fig. 6A), demonstrating enhanced tolerance to both random and targeted disruptions. In contrast, the ORSF network showed stronger dependency on a limited set of core topological elements, increasing its vulnerability to structural disturbance.

Analysis of keystone species identified a greater number of keystone taxa in the ORSF network (Fig. 6B and C). While both networks were dominated by Ascomycota and Basidiomycota, these phyla occupied distinct ecological niches: most ascomycetes functioned as peripheral nodes with high functional redundancy, whereas basidiomycetes, characterized by significantly higher within-module connectivity ($P < 0.01$), primarily served as module hubs.

Topological comparison revealed fundamental structural differences between the networks (Fig. 6D; Table S1). The OMF network featured fewer nodes but a significantly higher edge-to-node ratio ($P < 0.001$), indicating denser interconnections. Conversely, the ORSF network contained more nodes but sparser connectivity. The integrated OMF-ORSF cross-domain network displayed the tightest node connections and highest average degree, effectively combining structural advantages from both compartments to form a robust ecological system.

Geographical analysis of subnetworks showed clear population-level differentiation in complexity. GDHY and GDCZ formed high-complexity networks; GDZH, GDMM, and GDHZ exhibited intermediate complexity, while GDZS, GDSZ, and FJYX maintained

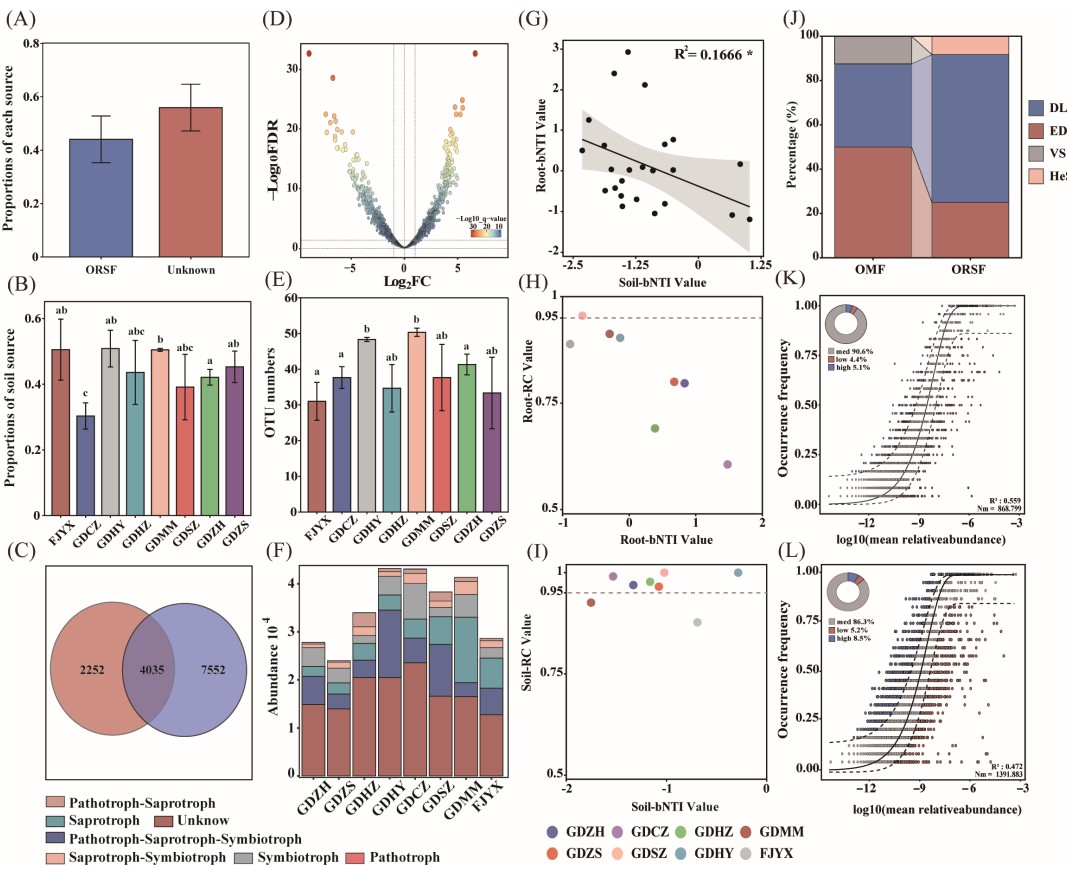

**FIG 5** Soil-derived fungal recruitment and community assembly processes in the root microbiome. (A and B) Soil contribution to root mycobiota: (A) SourceTracker analysis reveals that soil serves as the primary reservoir for root-associated fungi, contributing an average of 44.05% of root fungal communities. (B) The proportion of soil-derived fungi shows significant variation among populations, indicating differential recruitment across geographical gradients. (C–F) Host-selective enrichment patterns: (C) Venn diagram illustrates substantial OTU overlap between root and rhizosphere compartments, with roots harboring a specialized subset of the soil fungal pool. (D) Differential abundance analysis identifies OTUs significantly enriched in roots compared to rhizosphere soil. (E) Population-level variation in the number of root-enriched OTUs highlights differential host selection across sites. (F) Functional profiling shows that root-enriched OTUs are dominated by fungi with unknown trophic modes, suggesting specialized, potentially orchid-specific functions. (G–J) Deterministic versus stochastic assembly processes: (G) Significant negative correlation between root and rhizosphere fungal community βNTI values indicates contrasting assembly drivers between the two compartments. (H and I) βNTI–RC Bray ternary plots demonstrate that root-associated fungi (H) are primarily governed by ecological drift, while rhizosphere soil fungi (I) are dominated by dispersal limitation. (J) Null model quantification confirms the predominance of ecological drift in roots versus dispersal limitation in soil across all studied populations. (K and L) Neutral model fit: (K and L) Sloan neutral model predictions show stronger fit for rhizosphere soil fungi ($R^2 = 0.443$) than for root-associated fungi ($R^2 = 0.286$), indicating that stochastic processes play a greater role in soil community assembly, while host selection introduces deterministic filtering in root communities.

medium-low complexity networks, suggesting that local environmental conditions and host characteristics collectively shape network architecture in *P. purpuratum*.

In all network diagrams, node color represents fungal phyla (Ascomycota, Basidiomycota, Chytridiomycota, Glomeromycota, and others). Node size is proportional to the node degree, a key topological parameter indicating the number of connections. Edges depict significant positive or negative correlations between nodes, with their thickness corresponding to the strength of the relationship.

## DISCUSSION

### Mycorrhizal specificity characteristics and regulatory mechanisms of environmental factors

The specificity of orchid mycorrhizal associations is governed by a complex interplay of nutritional mode, ecological habit, and geographical distribution (31, 32). *P. purpuratum*,

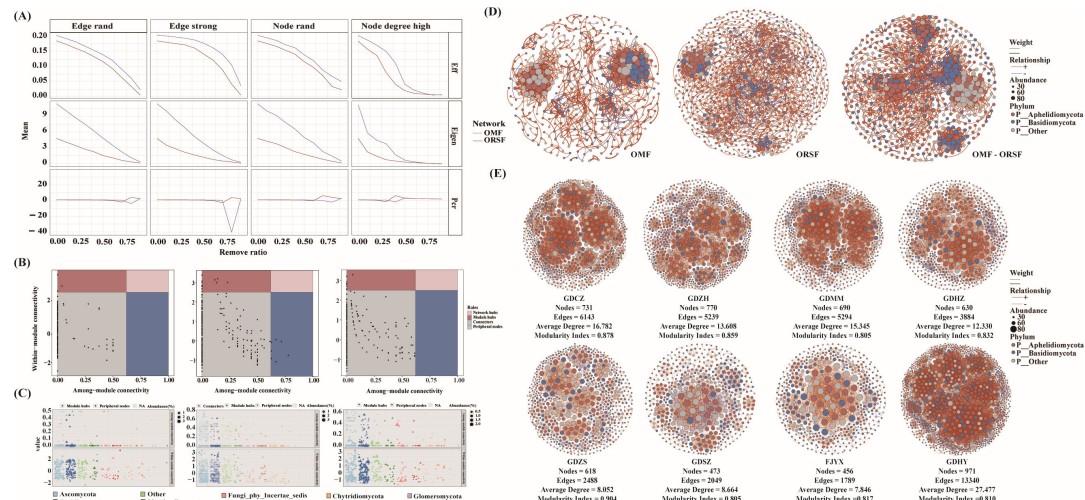

**FIG 6** Topological features and stability of fungal co-occurrence networks across compartments and populations. (A) Network robustness assessment under progressive node removal demonstrates that the OMF network maintains higher efficiency (Eff) and eigenvector centrality than the other root-associated soil fungi (ORSF) network, indicating superior stability and resistance to structural disturbance. (B and C) Identification of keystone taxa: (B) Z–P plot classifies nodes by within-module connectivity ($Z_i$) and among-module connectivity ($P_i$), revealing that ORSF contains more keystone species. (C) Phylum-level distribution shows that Ascomycota primarily function as peripheral nodes with high functional redundancy, while Basidiomycota, characterized by high within-module connectivity, frequently serve as module hubs. (D) Global network architecture visualization reveals distinct topological structures: the OMF network exhibits dense interconnections (high edge/node ratio), while the ORSF network shows larger size but sparser connectivity. The integrated OMF–ORSF network demonstrates enhanced connectivity and average degree, forming a robust ecological functional system. (E) Population-level subnetwork complexity displays clear geographical differentiation, with GDHY and GDCZ forming high-complexity networks, GDZH/GDMM/GDHZ showing intermediate complexity, and GDZS/GDSZ/FJYX maintaining medium-low complexity networks.

with its functional green leaves and capacity for photosynthetic carbon assimilation, represents a photosynthetic autotroph with relatively low fungal dependency. This nutritional strategy aligns with its observed low mycorrhizal specificity, as evidenced by significant compositional variation in root-associated fungi across taxonomic levels (phyla to genera). This pattern mirrors findings in other photosynthetic orchids including *Gavilea australis* (33), *Liparis japonica* (34), and *Dendrobium* spp. (35), contrasting sharply with the high specificity observed in non-photosynthetic taxa like *Corallorhiza striata*, which demonstrates specialized associations with *Thelephoraceae* fungi (36).

As a terrestrial orchid, *P. purpuratum* exhibits potential functional specificity during seed germination, relying on *Tulasnellaceae* and *Sebacinales* fungi for carbon provision. However, this specificity appears more flexible than in lithophytic, epiphytic, or saprophytic orchids, which typically show stronger fungal preferences due to more constrained environmental conditions (37). Geographically, the predominantly subtropical distribution of *P. purpuratum* across southern China and northern Vietnam further supports its low specificity, consistent with the general pattern of reduced mycorrhizal specialization in tropical and subtropical orchids (32). This stands in contrast to the high specificity observed in island-endemic species like *Anoectochilus sandvicensis* (38) and highlights the role of geographic isolation in driving specialization, as demonstrated by the divergent fungal associations of *Gymnadenia conopsea* and *Epipactis helleborine* between European and Asian populations (39).

Our multivariate analyses (PCA and dbRDA) confirmed significant population-level differentiation in fungal communities, primarily driven by abiotic factors that collectively explained 30.52% of the observed variation. This strong environmental filtering effect, operating under weak functional constraints, promotes compositional divergence among populations (40). Longitude emerged as the dominant spatial factor, likely serving as a proxy for altitudinal and thermal gradients that collectively shape orchid distribution and fungal community structure (41). Random forest modeling identified

solar radiation and precipitation as key environmental determinants. Experimental evidence suggests that temperature increases and precipitation alterations significantly impact fungal biomass and community structure, with warming and reduced rainfall particularly enhancing fungal phospholipid fatty acid content and potential soil respiration rates (42).

Mycorrhizal symbiosis further enhances plant adaptation to high-light environments through improved photosynthetic performance and photoprotective mechanisms (43). The critical role of light regimes and geological substrates in regulating orchid diversity has been confirmed across multiple species (41). Precipitation mediates plant-fungal relationships through soil moisture regulation, as demonstrated in *Platanthera grandiflora* where increased rainfall elevated non-mycorrhizal fungal proportions (44). Precipitation further influences arbuscular mycorrhizal fungal communities via pH modification (45). Additional factors including soil moisture (46), nutrient availability (47), and temporal variation (48) may further modulate specificity patterns.

In conclusion, *P. purpuratum* exemplifies how photosynthetic autotrophy, terrestrial habit, and subtropical distribution converge to maintain low mycorrhizal specificity, while environmental factors—especially solar radiation and precipitation—act as primary drivers of fungal community assembly across its geographical range. This integrated perspective advances our understanding of orchid–fungal symbiosis ecology and provides a framework for predicting microbial responses to environmental change in threatened orchid species.

## Screening mechanisms and ecological significance of core fungal taxa

The identification of core fungal taxa in *P. purpuratum* roots employed two complementary screening strategies that revealed fundamentally different aspects of community organization: spatial persistence-based screening and interaction-based screening.

## Spatial persistence and interaction-based screening

The Venn diagram approach identified 355 persistent OTUs that maintained stable presence across all geographic populations, selecting for taxa characterized by exceptional environmental adaptability and consistent host association. In contrast, network analysis prioritized taxa based on their connectivity within the microbial community, identifying 373 keystone OTUs with node degree >1, emphasizing biological interactions rather than mere presence/absence.

The dominant families identified through persistence-based screening—*Herpotrichiellaceae* and *Nectriaceae*—demonstrate remarkable ecological plasticity. It is important to note that while *Herpotrichiellaceae* members have been documented as functional symbionts in various plant systems (49) and their ecological success is hypothesized to be aided by physiological adaptations such as melanin-rich cell walls (50), these functional attributions based on phylogenetic inference require further experimental validation. Similarly, the widespread distribution of *Nectriaceae* fungi as root endophytes across diverse orchid species (51), along with genomic analyzes revealing expanded genetic repertoires (52), suggests, but does not conclusively prove, a capacity for diverse ecological interactions within our system.

## Convergent ecological patterns and functional specialization

Notably, both screening approaches consistently identified three families—*Nectriaceae*, *Russulaceae*, and *Thelephoraceae*—highlighting their dual importance in both environmental persistence and interactive roles. *Russulaceae* and *Thelephoraceae* represent highly diverse ectomycorrhizal lineages with broad geographic distributions (53), particularly abundant in subtropical and tropical regions (54). Their simultaneous identification through both methods suggests these families successfully balance environmental adaptability with specialized host interactions, with *Thelephoraceae* particularly noted for its abundance and diversity across latitudinal gradients (55).

The functional specialization of these screening approaches was particularly evident for OMF, which showed significantly greater abundance in interaction-defined keystone OTUs compared to persistent OTUs. This distribution pattern reflects the sophisticated molecular dialog between orchids and their mycorrhizal partners (56), wherein hosts employ precise signaling mechanisms to selectively enrich beneficial fungi through hormone/flavonoid-mediated stimulation and receptor-protein recognition systems that trigger mycorrhizal symbiosis-related gene expression (57). Crucially, the stark contrast in OMF abundance between the two screening methods underscores that a stable spatial presence alone is a poor predictor of functional importance in symbiotic interactions; host-mediated biological filtering, as captured by network analysis, is a more definitive indicator of symbiotic significance.

These complementary screening frameworks collectively demonstrate that core fungal assemblages in *P. purpuratum* roots are shaped by the interplay between environmental filtering (selecting for persistent taxa) and host-mediated biological filtering (selecting for interactive taxa). The convergent identification of specific families highlights taxa that successfully integrate environmental adaptability with specialized host interactions, representing the ecological and functional core of the orchid mycobiome.

## Recruitment mechanisms and community assembly processes of root-associated fungi

Our findings demonstrate that the assembly of root-associated fungal communities in *P. purpuratum* is governed by complementary ecological processes operating across different spatial and temporal scales. The identification of 355 persistent OTUs shared across all populations, dominated by *Herpotrichiellaceae* and *Nectriaceae*, reveals a core mycobiome characterized by remarkable environmental adaptability. These families have been widely documented as functional symbionts in various plant systems, with *Herpotrichiellaceae* exhibiting physiological adaptations such as melanin-rich cell walls that confer stress tolerance (50).

Plant roots actively shape the rhizosphere microenvironment by releasing exudates such as organic acids, sugars, amino acids, and secondary metabolites, thereby facilitating symbiotic relationships with mycorrhizal fungi (58, 59). This active rhizosphere engineering enables plants to selectively recruit soil fungi according to their physiological needs and environmental conditions, as demonstrated by the preferential colonization of specific fungal taxa in various plant systems under stress conditions (60, 61).

Supporting the concept of soil as a key species reservoir, Ballauff (62) reported 56% of root-associated fungal OTUs in a Sumatran ecosystem were shared with surrounding soil. Our finding that approximately 44% of root-associated fungi in *P. purpuratum* were soil-derived reinforces this concept while highlighting the selective nature of host recruitment. Among the 235 root-enriched OTUs, the functional profile reveals intriguing ecological strategies: nearly half (49.61%) were of unknown trophic type, indicating substantial knowledge gaps in fungal functional ecology, while multi-trophic taxa (18.19%) including *Ceratobasidium* (63) and *Fusarium* (64) suggest ecological flexibility as an adaptive strategy. The minimal presence of pathogenic fungi (<0.1%) demonstrates effective host-mediated filtering, while the enrichment of ectomycorrhizal taxa like *Russula* and *Tomentella* indicates active recruitment of beneficial symbionts.

Community assembly processes exhibit clear compartmentalization between root and soil habitats. The rhizosphere fungal community was primarily governed by dispersal limitation, consistent with patterns observed in various soil microbial systems (65, 66). In contrast, root-associated fungi were predominantly structured by ecological drift, mirroring findings in other plant systems including pear trees (67), creating a selectively permeable barrier that filters incoming soil fungi. We propose a conceptual model wherein strong host filtering acts as a primary deterministic sieve, creating a phylogenetically and functionally narrowed pool of colonists. Once inside the root, these colonists inhabit a milieu of reduced interspecific competition and environmental fluctuation,

which in turn amplifies the relative influence of stochastic ecological drift through random birth–death events and local extinctions (68).

The prevalence of putative multi-trophic fungi, operating within a drift-dominated assembly framework, may represent an effective ecological strategy for the host. This configuration—a "portfolio" of functionally versatile but stochastically assembled symbionts—could enhance host resilience to environmental variability. Even as species composition fluctuates randomly due to drift, the persistence of multi-trophic guilds ensures that critical ecosystem services (e.g., saprotrophy, symbiosis) are maintained, thereby conferring functional stability amidst taxonomic stochasticity. While environmental filtering undoubtedly shapes the available species pool, host regulation and stochastic processes interact synergistically to structure the final root mycobiome, creating a system that balances specificity with adaptive capacity—a characteristic particularly advantageous for long-lived perennial species in heterogeneous environments.

## Co-occurrence networks of root-associated and rhizosphere fungal communities

Microbial network analysis reveals fundamental differences in topological organization between OMF and ORSF fungal communities in *P. purpuratum*. The OMF network demonstrated superior stability metrics, exhibiting higher global efficiency and eigenvector centrality that reflect enhanced connectivity and regulatory capacity of key nodes. This "compact and interconnected" architecture—characterized by fewer nodes but higher edge-to-node ratio—aligns with the stable root internal environment and host-mediated symbiotic selection pressure (69). The tightly connected symbiotic modules not only improve nutrient acquisition efficiency but also strengthen network resilience against disturbance, forming an adaptive architecture where mycorrhizal fungi serve as key nodes within a core-periphery structure reinforced by directional recruitment through root exudates (70).

In contrast, the ORSF network displayed lower robustness with a "node-rich but loosely connected" pattern. This structure, defined by higher node numbers but lower edge-to-node ratio, represents an adaptive strategy to the frequently fluctuating rhizosphere environment. While this configuration provides functional flexibility through redundant nodes and accommodates diverse keystone species to manage environmental complexity, it becomes particularly vulnerable to collapse under extreme conditions (71). The extensive but non-specific species interactions in the rhizosphere follow the "habitat complexity-keystone species number" correlation principle, trading redundant nodes for responsiveness to resource variations.

The integrated OMF-ORSF network effectively combined the stability of root-associated fungi with the functional breadth of rhizosphere communities, exhibiting the highest average degree and connection density to form an efficient and stable cross-habitat system. This integration creates a robust functional framework that leverages complementary strengths from both compartments.

Geographical analysis revealed substantial population-level variation in network complexity, with GDHY constructing the most complex network characterized by highest node count, edge number, and average degree. This "high richness-high interaction density" architecture provides multiple functional advantages: redundant pathways can compensate for damaged connections, high average degree enhances coordination efficiency in nutrient cycling, and diverse nodes undertake specialized tasks including organic matter decomposition and pathogen suppression (72). For such populations, maintaining habitat complexity and heterogeneity is crucial to prevent network degradation. Conversely, GDZS and FJYX exhibited reduced network complexity, with FJYX showing characteristics potentially indicative of microbial diversity loss due to habitat fragmentation. The high modularity in GDZS suggests functional specialization as an adaptive strategy to habitat constraints, potentially enhancing tolerance to extreme environments despite reduced overall diversity (73). FJYX minimal network

metrics—lowest node count, edge number, and average degree—resemble network degradation patterns observed in fragmented ecosystems (74), representing a potential indicator of population degradation that requires urgent habitat restoration to rebuild microbial interaction networks.

These network architectures reflect the complex interplay between habitat stability and host selection pressure, while providing practical insights for developing targeted conservation strategies that maintain microbial community integrity in endangered orchid species.

## Conclusions

This study demonstrates that the root-associated fungal communities of *P. purpuratum* are characterized by low mycorrhizal specificity and marked geographical differentiation, driven primarily by abiotic environmental factors. Through complementary screening approaches, we identified distinct ecological strategies among core taxa: spatially persistent fungi exhibited broad environmental adaptability, whereas interaction-based keystone taxa were disproportionately enriched in symbiotic fungi, particularly OMF occupying critical network positions. Stochastic processes dominated community assembly, with ecological drift prevailing in root-associated communities and dispersal limitation shaping rhizosphere assemblages. These findings provide mechanistic insights into the ecological forces structuring orchid–fungal symbioses and establish a multidisciplinary framework to support the targeted conservation of this endangered orchid species.

### ACKNOWLEDGMENTS

This work was supported by the Self-deployed Basic Research Project of South China Botanical Garden, Chinese Academy of Sciences (*P. purpuratum*) (grant number JCYJXM-2025012) and the Guangdong Science and Technology Plan Project (grant number 2022B1111230001).

Y.T.: conceptualization, methodology, data curation, investigation, and writing— original draft; J.L.: investigation and validation; Q.Y.: reviewing, editing, funding acquisition, supervision, and project management.

### AUTHOR AFFILIATIONS

[1]State Key Laboratory of Plant Diversity and Specialty Crops, Key Laboratory of National Forestry and Grassland Administration on Plant Conservation and Utilization in Southern China, Guangdong Provincial Key Laboratory of Applied Botany, Chinese Academy of Sciences South China Botanical Garden, Guangzhou, Guangdong, China
[2]South China National Botanical Garden, Guangzhou, China.
[3]College of Agriculture and Biology, Zhongkai University of Agriculture and Engineering, Guangzhou, China

### AUTHOR ORCIDs

Qifei Yi http://orcid.org/0000-0002-1669-3265

### AUTHOR CONTRIBUTIONS

Yong Tan, Conceptualization, Data curation, Investigation, Methodology, Validation, Visualization, Writing – original draft | Junxi Liang, Investigation, Validation | Qifei Yi, Conceptualization, Funding acquisition, Project administration, Supervision, Writing – review and editing

### ADDITIONAL FILES

The following material is available online.

## Supplemental Material

**GRAPHICAL ABSTRACT (Spectrum02573-25-s0001.tif).** For *Paphiopedilum purpuratum*, geographic environmental factors drive root mycobiota differentiation, while host selection (coupled with ecological drift and dispersal limitation) shapes its rhizosphere fungal community assembly—ultimately forming a robust, efficient OMF-ORSF network system via environment-host synergy.

**Table S1 (Spectrum02573-25-s0002.docx).** Indices of the topological structure of the microbial community network.

## Open Peer Review

**PEER REVIEW HISTORY (review-history.pdf).** An accounting of the reviewer comments and feedback.

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
