## [Reviewer comments · Microbiology Spectrum]

Microbiology Spectrum

Study on geographic differentiation and environment-host synergistic assembly mechanism of root-associated fungal communities in *Paphiopedilum purpuratum*

Tan Yong, Liang Junxi, and Yi Qifei

Corresponding Author(s): Yi Qifei, Chinese Academy of Sciences South China Botanical Garden

Review Timeline:

Submission Date:	August 22, 2025
Editorial Decision:	October 7, 2025
Revision Received:	October 27, 2025
Accepted:	December 11, 2025

Editor: Weimin Sun

Reviewer(s): Disclosure of reviewer identity is with reference to reviewer comments included in decision letter(s). The following individuals involved in review of your submission have agreed to reveal their identity: Youwei Zuo (Reviewer #1)

Transaction Report:

DOI: <https://doi.org/10.1128/spectrum.02573-25>

Re: Spectrum02573-25 (Study on geographic differentiation and environment-host synergistic assembly mechanism of root-associated fungal communities in *Paphiopedilum purpuratum*)

Dear Dr. Yi Qifei:

Thank you for the privilege of reviewing your work. Below you will find my comments, instructions from the Spectrum editorial office, and the reviewer comments.

Revision Guidelines

Sincerely,
Weimin Sun
Editor
Microbiology Spectrum

Reviewer #1 (Comments for the Author):

This study systematically investigated the composition, diversity, environmental drivers, and community assembly mechanisms of the *Paphiopedilum purpuratum* root-associated.

1. The relationship between node color, size, and topological parameters in the network diagram (Figure 7) should be clearly stated in the figure caption to avoid reader misinterpretation.

2. In Figures 2-5, some figure captions are insufficiently detailed. It is recommended to supplement statistical test indicators (such as P-values, letters of difference) and clearly specify the test methods used in the Methods section.

3. Some sentences are excessively long; it is recommended to appropriately split them to enhance readability.

4. Currently, each population has only 3 biological replicates, which, although conventional, may be insufficient for the study of geographic differentiation. It is recommended to include a justification for the sample size in the discussion or methods section. This study features rich data, advanced analytical approaches, and convincing conclusions, holding significant ecological and conservation biology implications. The findings align well with *Microbiology Spectrum's* focus on mechanism-driven research in microbial ecology. It is recommended that the manuscript undergo moderate revisions prior to acceptance, focusing on the following areas: clarifying sample size justification, interpreting functional predictions with caution, providing a deeper discussion of community assembly mechanisms, and optimizing the figures and tables.

Reviewer #2 (Comments for the Author):

This is an interesting and relevant manuscript that examines root-associated fungal communities in *Paphiopedilum purpuratum* across multiple wild populations. The dataset and the ecological framing have merit, but the paper requires substantial editorial polishing and several methodological clarifications before it can be considered for publication.

1. Language and formatting. The manuscript contains numerous typographical issues (e.g., words accidentally joined) and even residual non-English characters in the Methods (for instance, a Chinese term appears). Please perform a thorough English edit and remove non-English artifacts.

2. Bacteria-fungi data handling. You report merging bacterial 16S and fungal ITS datasets within a single phyloseq object and conducting cross-domain analyses. Please justify this choice. This will help readers interpret domain-specific patterns prior to cross-domain inference.

3. Methods transparency and citations. Several procedures, packages, and statistics are named without enough explanation or primary references.

4. Acronyms and units. Ensure all abbreviations are expanded at first mention in the main text and in figure legends. This includes environmental variables and biotic metrics (e.g., AMT, AI, ET, INT). Keep units consistent throughout.

5. Taxonomic formatting. Genus and species names must be italicized consistently throughout the text and figures/captions.

6. Sampling metadata. You state there were three biological replicates per site across eight sites; however, key spatial metadata are missing. Please add GPS coordinates for each site, describe the spatial layout of plant/soil subsampling (e.g., distance between sampled plants/cores), and clarify whether technical replicates were used.

7. Figure and table legends. Make the legends more self-contained and explanatory.

Dear Dr. Weimin Sun:

We are very grateful to your and the reviewers' critical comments and thoughtful suggestions. Based on these comments and suggestions, we have made careful modification on the original manuscript. Once again, we acknowledge your comments and constructive suggestions very much, which are valuable in improving the quality of our manuscript. Furthermore, we would like to show the details as follows:

Reviewer 1#

We sincerely thank Reviewer 1 for their thoughtful feedback and valuable suggestions. The comments have been instrumental in refining our work. Our responses to each specific point are as follows:

Comment 1: The relationship between node color, size, and topological parameters in the network diagram (Figure 7) should be clearly stated in the figure caption to avoid reader misinterpretation.

Response:

We thank the reviewer for this valuable suggestion. In response, we have revised the caption of Figure 7 to explicitly state the relationship between visual elements and their corresponding topological parameters. The updated caption now reads: "In all network diagrams, node color represents fungal phyla (Ascomycota, Basidiomycota, Chytridiomycota, Glomeromycota, and Others). Node size is proportional to the node degree, a key topological parameter indicating the number of connections. Edges depict significant positive or negative correlations between nodes, with their thickness corresponding to the strength of the relationship." We believe this clarification effectively prevents potential misinterpretation and thank the reviewer for helping us improve the clarity of our figure.

Comment 2: In Figures 2-5, some figure captions are insufficiently detailed. It is recommended to supplement statistical test indicators (such as P-values, letters of difference) and clearly specify the test methods used in the Methods section.

Response:

We sincerely thank the reviewer for this constructive suggestion. In response, we have thoroughly revised the captions of Figures 2–5 to include detailed statistical information. Specifically, we have now explicitly stated the statistical test methods used (one-way ANOVA with Tukey's post hoc test), significance levels ($P < 0.05$), and the meaning of the lowercase letters indicating significant differences between groups.

For example, the updated caption for Figure 2 now reads: "Fig. 2 Taxonomic composition and α -diversity of root-associated fungal communities across populations. (A–C) Relative abundance of fungal taxa at the phylum (A), family (B), and genus (C) levels. (D–F) Boxplots of α -diversity indices: Shannon (D), Simpson (E), and Chao1 (F). Significant differences among populations were assessed by one-way ANOVA with Tukey's post hoc test ($P < 0.05$). Groups not sharing the same lowercase letter differ significantly." In addition, as suggested, we have also clearly described the statistical methodology in the Methods section under "Statistical Analysis" to ensure full transparency and reproducibility. We believe these revisions have significantly improved the clarity and rigor of the figures and text, and we thank the reviewer again for the insightful comments.

Comment 3: Some sentences are excessively long; it is recommended to appropriately split them to enhance readability.

Response:

We thank the reviewer for this valuable feedback. We have thoroughly reviewed the manuscript and revised sentences that were excessively long or complex. Specifically, we have: Split lengthy sentences with multiple clauses into shorter, more focused sentences; Simplified complex grammatical structures; Ensured each sentence conveys a single clear idea; Maintained proper academic tone while improving readability. These edits have been applied throughout the manuscript to enhance overall readability while preserving all scientific content and precision.

Comment 4: Currently, each population has only 3 biological replicates, which,

although conventional, may be insufficient for the study of geographic differentiation. It is recommended to include a justification for the sample size in the discussion or methods section.

Response:

We thank the reviewer for raising this important point regarding sample size. We acknowledge that three biological replicates per population, while consistent with standard practice in ecological microbiome studies for capturing major community trends, may have limitations in resolving fine-scale geographic patterns.

In response to this comment, we have revised the Methods section to include an explicit justification for our sampling design. Specifically, we have added the following statement: "The sample size of three biological replicates per population was determined based on (1) the limited availability of representative individuals across the geographically distinct populations of this endangered species, and (2) the standard practice in ecological studies for capturing dominant community patterns. While this design provides robust initial insights into assembly mechanisms along geographical gradients, we acknowledge that future studies with expanded sampling would be valuable to resolve more subtle population-level differences."

These revisions provide transparency about our sampling rationale while appropriately framing the scope and limitations of our conclusions.

Overall Comments: This study features rich data, advanced analytical approaches, and convincing conclusions, holding significant ecological and conservation biology implications. The findings align well with Microbiology Spectrum's focus on mechanism-driven research in microbial ecology. It is recommended that the manuscript undergo moderate revisions prior to acceptance, focusing on the following areas: clarifying sample size justification, interpreting functional predictions with caution, providing a deeper discussion of community assembly mechanisms, and optimizing the figures and tables.

Response:

Thank you for your valuable comments. We have revised the Discussion section

accordingly to address your suggestions, placing greater emphasis on cautious interpretation of functional predictions and providing a more in-depth discussion of community assembly mechanisms. The main revisions include:

Regarding the cautious interpretation of functional predictions: In the "Core fungal taxa" section, we have incorporated more qualifying language (e.g., "hypothesized," "suggest, but does not conclusively prove"). We explicitly state that the understanding of the functions of families like Herpotrichiellaceae and Nectriaceae is largely based on phylogenetic inference and emphasize that their precise ecological roles require further experimental validation. We specifically added a discussion highlighting that "spatial persistence" does not equate to "functional importance," thereby underscoring the superiority and necessity of using interaction-based network analysis to identify functionally core taxa.

Regarding the in-depth discussion of community assembly mechanisms: We propose a more integrated "Filter-Drift" conceptual model. This model clearly explains how deterministic processes (host filtering) create the conditions for stochastic processes (ecological drift) to dominate the assembly of the root-endophytic fungal community. This deepens the mechanistic understanding of how different ecological processes operate sequentially and in a compartment-specific manner. We further elaborated on the ecological advantages of maintaining "multi-trophic taxa" within a "stochastically assembled" framework. By introducing the "functional portfolio" concept, we explain the mechanism through which critical ecosystem functions can be maintained even as species composition fluctuates randomly. This more tightly links community assembly processes with host plant adaptability.

We believe these revisions adequately address your concerns and significantly enhance the theoretical depth and scientific rigor of the manuscript. Thank you again for your insightful comments.

In addition to the specific comments addressed above, we have thoroughly proofread the manuscript to improve linguistic clarity and consistency. These revisions include standardizing the reference format and adding a new graphical

abstract.

Reviewer 2#

We sincerely thank Reviewer 2 for their thoughtful feedback and valuable suggestions. The comments have been instrumental in refining our work. Our responses to each specific point are as follows:

Comment 1: Language and formatting. The manuscript contains numerous typographical issues (e.g., words accidentally joined) and even residual non-English characters in the Methods (for instance, a Chinese term appears). Please perform a thorough English edit and remove non-English artifacts.

Response:

We thank the reviewer for highlighting these important language and formatting issues. We have now performed a thorough English edit of the entire manuscript with the assistance of a professional editing service to address typographical errors, grammatical inaccuracies, and formatting inconsistencies. All non-English characters, including the Chinese term noted in the Methods section, have been carefully identified and removed. Additionally, we have standardized the reference format throughout the manuscript to ensure full consistency with journal guidelines. We believe these comprehensive revisions have significantly improved the clarity and professionalism of the manuscript.

Comment 2: Bacteria-fungi data handling. You report merging bacterial 16S and fungal ITS datasets within a single phyloseq object and conducting cross-domain analyses. Please justify this choice. This will help readers interpret domain-specific patterns prior to cross-domain inference.

Response:

We thank the reviewer for this insightful comment regarding our cross-domain analytical approach. The integration of bacterial 16S rRNA gene and fungal ITS datasets into a unified phyloseq object was motivated by key ecological considerations rooted in our findings. Specifically, our results showed that

approximately 44% of root-associated fungi were derived from the rhizosphere soil, establishing a direct microbial linkage between these compartments.

This integrated approach is ecologically justified for two primary reasons. First, rhizosphere fungal and bacterial communities concurrently respond to environmental fluctuations and collectively influence plant physiology and health. Second, complex interkingdom interactions among rhizosphere microbes can subsequently affect the composition and function of root-associated fungal communities through both direct microbial interactions and plant-mediated mechanisms. Therefore, cross-domain analysis provides a necessary framework to reveal the complex interkingdom interactions that shape root mycobiome assembly.

Comment 3: Methods transparency and citations. Several procedures, packages, and statistics are named without enough explanation or primary references.

Response:

We thank the reviewer for highlighting the need for greater methodological transparency. In response, we have thoroughly revised the Methods section to provide complete citations for all R packages and analytical procedures, along with detailed explanations of key statistical approaches. Specifically:

All R packages now include proper version numbers and primary references (e.g., phyloseq v1.52.0 (21), vegan v2.6-10 (22)). Key analytical procedures now include both methodological references and implementation details: Neutral community model: neutralCommunityModel package with R^2 and Nm parameters (28); Network analysis: SparCC algorithm with correlation ≥ 0.6 and $P < 0.05$ thresholds (28); Statistical thresholds and criteria are explicitly stated throughout (e.g., $P < 0.05$, $|\log_2FC| \geq 1$).

All function names and parameters are clearly specified to ensure reproducibility. These revisions provide complete methodological transparency and ensure proper attribution of all analytical approaches.

Comment 4: Acronyms and units. Ensure all abbreviations are expanded at first

mention in the main text and in figure legends. This includes environmental variables and biotic metrics (e.g., AMT, AI, ET, INT). Keep units consistent throughout.

Response:

We thank the reviewer for the valuable feedback regarding acronyms and units. We have thoroughly revised the manuscript to address these concerns as follows:

1. Expanded all acronyms at first mention: All environmental variables now include full names with abbreviations in parentheses (e.g., "annual mean temperature (AMT)"), All biotic metrics are clearly defined with descriptive names;
 2. Standardized units throughout: Soil properties: used consistent units (g/kg for total elements, mg/kg for available nutrients), Climate variables: applied standardized unit formats (e.g., m/s for wind speed, kJ/m²/day for solar radiation), Maintained consistent unit presentation format (value, space, unit);
 3. Improved clarity of biotic metrics: Used more descriptive names for network topology metrics, Maintained consistent terminology across the manuscript;
 4. Ensured consistency across all sections: Applied the same standards to main text, figure legends, and tables, Verified unit consistency in all methodological descriptions
- These revisions ensure that all abbreviations are properly introduced and units are consistently presented throughout the manuscript, greatly improving readability and methodological clarity.

Comment 5: Taxonomic formatting. Genus and species names must be italicized consistently throughout the text and figures/captions.

Response:

We thank the reviewer for highlighting this important formatting requirement. We have now carefully reviewed the entire manuscript, including all figures and captions, to ensure consistent italicization of all genus and species names throughout the text. All scientific names have been formatted according to standard taxonomic conventions.

Comment 6: Sampling metadata. You state there were three biological replicates per site across eight sites; however, key spatial metadata are missing. Please add GPS coordinates for each site, describe the spatial layout of plant/soil subsampling (e.g., distance between sampled plants/cores), and clarify whether technical replicates were used.

Response:

We thank the reviewer for this important comment regarding sampling metadata. We understand the need for comprehensive spatial documentation and have taken the following measures to address this concern while balancing conservation requirements:

1. Spatial Reference: As indicated in Figure 1, we have provided the approximate geographic distribution of all eight sampling populations. However, due to the endangered status of *Paphiopedilum purpuratum* and severe poaching pressures in the region, we are unable to publish exact GPS coordinates to protect the remaining wild populations.
2. Sampling Design Clarification: We have expanded the Methods section to provide detailed description of our sampling strategy: Sampling was conducted in typical areas with relatively high plant density within each population; Healthy plants were randomly selected using sterile gloves; For low-density populations, sampling intervals were maintained at >50 cm; For high-density populations, sampling intervals were increased to >1 m to ensure spatial independence; Root samples from 2-3 individual plants were pooled to form one biological replicate; Each site included three biological replicates for both root and rhizosphere soil samples;
3. Technical Replicates: We confirm that no technical replicates were used in this study. Each biological replicate represents an independent sampling unit as described above.

We believe this sampling approach provides robust representation of each population while maintaining ethical conservation practices for this endangered species. The sampling intervals were designed to ensure spatial independence and capture population-level characteristics rather than microsite variations.

Comment 7: Figure and table legends. Make the legends more self-contained and explanatory.

Response:

We thank the reviewer for the valuable suggestion to improve our figure legends. We have thoroughly revised all figure legends to make them more self-contained and explanatory. Specifically, we have:

Added detailed methodological descriptions for each analytical approach;

Included key statistical results and interpretations directly in the legends;

Provided clear explanations of abbreviations and symbols used in the figures;

Expanded the ecological interpretations of the visualized patterns.

For example, in Figure 2, we now explicitly state the statistical methods used (one-way ANOVA with Tukey's post hoc test) and the meaning of the significance indicators. In Figure 6, we provide comprehensive explanations of network topology parameters and their ecological implications.

These improvements ensure that each legend now stands alone as an informative summary of the corresponding figure, providing readers with sufficient context to understand both the methodology and biological significance of each visualization without needing to constantly refer back to the main text.

Overall Comments: This is an interesting and relevant manuscript that examines root-associated fungal communities in *Paphiopedilum purpuratum* across multiple wild populations. The dataset and the ecological framing have merit, but the paper requires substantial editorial polishing and several methodological clarifications before it can be considered for publication.

Response:

We sincerely thank the reviewer for their positive assessment of our manuscript's ecological framework and dataset, and for their constructive feedback regarding editorial polish and methodological clarification. We have thoroughly addressed each of the points raised, as detailed below.

1. Editorial Polishing We have performed comprehensive language editing throughout the manuscript to improve clarity, consistency, and overall readability. This included: Correcting grammatical errors and typographical mistakes; Ensuring consistent terminology and formatting; Simplifying overly complex sentence structures; Verifying that all genus and species names are properly italicized.

2. Methodological Clarifications Key methodological details have been clarified or expanded in the revised manuscript, including: Justification for the integration of bacterial and fungal datasets in cross-domain analyses; Explicit citations and version numbers for all R packages and analytical procedures; Expanded description of sampling design and spatial layout; Clarification of statistical thresholds and criteria used in all analyses

We believe these revisions have significantly strengthened the manuscript and addressed the reviewer's concerns. We are grateful for the input that has helped us improve the quality and clarity of our work.

We sincerely appreciate the time and effort that the reviewers have invested in evaluating our manuscript. We are eager to receive any additional feedback or suggestions that may further enhance our work.

Yours sincerely,

Qifei Yi

October 19, 2025

South China National Botanical Garden

Re: Spectrum02573-25R1 (Study on geographic differentiation and environment-host synergistic assembly mechanism of root-associated fungal communities in *Paphiopedilum purpuratum*)

Dear Dr. Yi Qifei:

Although Reviewer #1 declined to review the manuscript, the authors have meticulously addressed Reviewer #1's questions. Consequently, I concur with Reviewer #2's recommendation, and this paper is acceptable for publication.

Your manuscript has been accepted, and I am forwarding it to the ASM production staff for publication. Your paper will first be checked to make sure all elements meet the technical requirements. ASM staff will contact you if anything needs to be revised before copyediting and production can begin. Otherwise, you will be notified when your proofs are ready to be viewed.

Sincerely,
Weimin Sun
Editor
Microbiology Spectrum

Reviewer #2 (Comments for the Author):

I have reviewed the revised version of the manuscript and the authors' responses to my previous comments. I am satisfied with the revisions made and consider that all my concerns have been adequately addressed. I have no further comments or suggestions.